# Metformin Suppresses Stemness of Non-Small-Cell Lung Cancer Induced by Paclitaxel through FOXO3a

**DOI:** 10.3390/ijms242316611

**Published:** 2023-11-22

**Authors:** Zhimin Tang, Yilan Zhang, Zhengyi Yu, Zhijun Luo

**Affiliations:** 1Jiangxi Provincial Key Laboratory of Tumor Pathogens and Molecular Pathology, Department of Pathophysiology, School of Basic Medical Sciences, Nanchang University, Nanchang 330031, China; 356400210010@email.ncu.edu.cn; 2Nanchang Joint Program, Queen Mary School, Nanchang University, Nanchang 330031, China; 4217120186@email.ncu.edu.cn (Y.Z.); ncuyzy@email.ncu.edu.cn (Z.Y.)

**Keywords:** metformin, paclitaxel (PTX), NSCLC, stemness, FOXO3a

## Abstract

Cancer stem cells (CSCs) play a pivotal role in drug resistance and metastasis. Among the key players, Forkhead box O3a (FOXO3a) acts as a tumor suppressor. This study aimed to unravel the role of FOXO3a in mediating the inhibitory effect of metformin on cancer stemness derived from paclitaxel (PTX)-resistant non-small-cell lung cancer (NSCLC) cells. We showed that CSC-like features were acquired by the chronic induction of resistance to PTX, concurrently with inactivation of FOXO3a. In line with this, knockdown of FOXO3a in PTX-sensitive cells led to changes toward stemness, while overexpression of FOXO3a in PTX-resistant cells mitigated stemness in vitro and remarkably curbed the tumorigenesis of NSCLC/PTX cells in vivo. Furthermore, metformin suppressed the self-renewal ability of PTX-resistant cells, reduced the expression of stemness-related markers (c-MYC, Oct4, Nanog and Notch), and upregulated FOXO3a, events concomitant with the activation of AMP-activated protein kinase (AMPK). All these changes were recapitulated by silencing FOXO3a in PTX-sensitive cells. Intriguingly, the introduction of the AMPK dominant negative mutant offset the inhibitory effect of metformin on the stemness of PTX-resistant cells. In addition, FOXO3a levels were elevated by the treatment of PTX-resistant cells with MK2206 (an Akt inhibitor) and U0126 (a MEK inhibitor). Collectively, our findings indicate that metformin exerts its effect on FOXO3a through the activation of AMPK and the inhibition of protein kinase B (Akt) and MAPK/extracellular signal-regulated kinase (MEK), culminating in the suppression of stemness in paclitaxel-resistant NSCLC cells.

## 1. Introduction

Lung cancer poses a serious threat to the health and lives of human beings, as it is still the leading cause of cancer deaths. Non-small-cell lung cancer (NSCLC) accounts for about 85% of all lung cancer cases. Patients with advanced NSCLC can only survive for 9–12 months. Chemotherapy is the main treatment in the advanced stages. Recently, immunotherapy, based on blockade of the PD-1/PD-L1 interaction alone or in combination with chemotherapy, has exhibited survival benefits [1]; however, most patients will eventually develop resistance, leading to metastasis and recurrence of the tumor. Hence, identifying the modulators of drug resistance and developing new drugs against them has been a focus in the treatment of lung cancer.

Paclitaxel (PTX), a plant derivative of Taxus pacificus, is a potent anticancer drug that is used for the treatment of human non-small-cell lung cancer, breast cancer, and ovarian cancer [2,3,4]. PTX binds to the tubulin β subunit and causes the stable polymerization of microbundles, disrupting the dynamic balance between microtubule polymerization and depolymerization and thus interfering with mitosis and inhibiting the proliferation of cancer cells [5]. PTX does improve the survival time of many cancer patients; however, it rapidly induces drug resistance, which is a major clinical barrier in the treatment of NSCLC. Drug resistance concurs with metastasis, wherein cancer stem cells (CSCs) play a crucial role. The mechanism by which CSCs are induced and regulated in NSCLC is complex and still incompletely elucidated.

The FOXO family are transcription factors that regulate a variety of cellular processes, including cell cycle progression, cell survival and metabolism [6]. The family consists of four members, FOXO1, FOXO3a, FOXO4, and FOXO6, among which FOXO1, 3a, and 4 are widely expressed in various tissues and organs, while FOXO6 is largely restricted to neural cells [7]. FOXO2 is a homologue of FOXO3a, and FOXO5 (FOXO3b) is only expressed in zebrafish [8]. The FOXO family is subjected to regulation by many signaling pathways. For example, Akt negatively regulates FOXO transcriptional activity through phosphorylation followed by binding to 14-3-3, which leads to sequestering it in the cytoplasm [9]. Oxidative stress upregulates FOXO through the activation of c-Jun N-terminal kinase (JNK) [10]. AMPK has been reported to phosphorylate and activate FOXOs [11]. In addition, other signaling pathways can regulate FOXO activity in cancer, depending on the tumor microenvironmental context [12,13].

FOXO3a plays an important role in drug resistance and CSC formation [14,15]. It has been shown that the inhibition of FOXO3a markedly facilitates gefitinib resistance and induces stem cell-like phenotypes of lung cancer cells. Conversely, overexpression of FOXO3a in gefitinib-resistant lung cancer cells effectively counteracts drug resistance [16]. Reduced FOXO3a expression, orchestrated by DNA methyltransferase 1 (DNMT1), emerges as a driving force for proliferation and tumorigenesis due to increasing the population of breast cancer stem cells [17]. In lung cancer and colorectal cancer, FOXO3a overexpression has been shown to repress stem cell-like features and tumor initiation and curtail the emergence of drug resistance [18].

Metformin, a biguanide chemical compound, is a first-line oral hypoglycemic medicine to treat type 2 diabetes. It reduces basal and postprandial blood glucose levels, lowers blood insulin content, and enhances insulin sensitivity [19]. Beyond its antidiabetic properties, epidemiological studies have shown that patients with type 2 diabetes who receive metformin have a significantly lower risk of developing cancer than those who do not [20,21]. Thereafter, a plethora of studies have reported that metformin exerts anticancer activity on many types of cancer [22,23,24] and has synergistic effects with multiple chemotherapeutic agents [25,26]. Although studies have revealed that metformin is more effective in inhibiting CSCs [25], its underlying molecular mechanism of inhibiting NSCLC stemness has not been fully elucidated [27]. One study delineated how metformin exerts its repressive impact on cancer cell proliferation and stemness in NSCLC by pinpointing Nemo-like kinase (NLK) as a candidate [28]. A separate investigation underscored the synergistic potential of metformin and salinomycin in eradicating cancer stem cells derived from NSCLC [29]. Additionally, a recent report illustrated that low doses of metformin prompt the nuclear localization of FOXO3a, leading to the downregulation of CSC markers. This concurs with reprogramming ovarian and breast cancer cells into non-cancerous states, a process likely orchestrated by the activation of AMPK [30]. Indeed, the literature has indicated that FOXO3a is phosphorylated by AMPK on at least three Ser residues: Ser413, Ser588 and Ser626 [31]. Despite these advances, the extent to which FOXO3a serves as a conduit for metformin’s efficacy in countering PTX-induced resistance remains an enigma.

Here, we unveil a notable interplay implicating the inactivation of FOXO3a and the concurrent emergence of stemness within PTX-resistant lung cancer cells. Intriguingly, metformin appears to be a key protagonist in this process. It effectively facilitates the nuclear localization of FOXO3a, thereby curbing the development of stem cell-like spheroids within 3D cultures. In conjecture, metformin reduces the expression of stemness markers in PTX-resistant cell lines through a FOXO3a-dependent pathway, which is mediated by AMPK. In addition, metformin prevents the abnormal activation of Akt and extracellular signal-regulated kinase (ERK) in PTX-resistant cells. Treatment of PTX-resistant cells with MK2206 (an Akt inhibitor) or U0126 (a MEK inhibitor) reactivates FOXO3a and restores the repression of the c-MYC oncogene. Collectively, our results suggest that FOXO3a inactivation is a linchpin in PTX resistance and that metformin targets multiple kinases, including AMPK, Akt and ERK, to dismantle their barriers by inhibiting FOXO3a, leading to the reversion of drug resistance.

## 2. Results

### 2.1. Establishment and Confirmation of PTX-Resistant Cell Lines

PTX-resistant cell lines, A549/PTX and H460/PTX, were established by stepwise drug induction. We determined the half maximal inhibitory concentration (IC_50_) values of paclitaxel in drug-resistant cells and their parental cell lines (Figure 1A). A significant increase in IC_50_ of PTX was observed in A549/PTX drug-resistant cells as compared with A549 cells (37.56 µM vs. 1.19 µM, 32-fold). Similarly, the IC_50_ of PTX in H460/PTX drug-resistant cells was 15-fold greater than that in H460 (1.78 µM vs. 0.12 µM) (Figure 1B). Then, we measured the rate of cell apoptosis using Annexin V and PI staining following treatment with PTX (Appendix A). Notably, PTX-resistant cells were more resistant to PTX-induced cytotoxicity compared to PTX-sensitive cells (1.59-fold vs. 0.95-fold) (Appendix A). In addition, both paclitaxel-resistant cell lines had a boosted ability to proliferate, in contrast with their parental cells (Figure 1C). Thus, these data indicated that A549 and H460 cells developed resistance upon chronic incubation with PTX.

To verify the stemness of PTX-resistant cells, parental cells and drug-resistant cells were suspended in ultra-low attachment culture dishes and allowed to form spheres. As shown in Figure 1D, the cells detached and formed small spheres from day 3. A549/PTX and H460/PTX drug-resistant cells had a greater ability to form spheres than their parental cells. The number of spheroids derived from A549/PTX and H460/PTX drug-resistant cells increased to 663 ± 122.4 (vs. 368 ± 62.9) and 489 ± 191.0 (vs. 218 ± 64.0) on day 5 (Figure 1E). Equally, the diameters of the spheres formed by A549/PTX and H460/PTX drug-resistant cells reached ~175 ± 31.8 µm (vs. 141 ± 22.4 µm) and ~263 ± 26.7 µm (vs. 153 ± 22.4 µm), respectively (Figure 1F). These observations suggest that drug resistance plays an important role in the induction of cancer stemness.

We next evaluated the effect of PTX and metformin on A549 and A549/PTX drug-resistant cells in two-dimensional (2D) or three-dimensional (3D) cultures. First, we assessed cell viability after the treatment of cells with PTX (20–320 nM) or metformin (0.1–10 mM) alone or with their combination. Metformin significantly inhibited A549/PTX drug-resistant cells and their sphere viabilities (Appendix A) and no significant changes were observed with PTX alone (Figure 1A and Appendix A). Furthermore, the metformin-induced reduction in cell viability of PTX-resistant cells was more pronounced than that of PTX-sensitive cells (metformin IC50 in 2D culture: 0.98 mM vs. 1.58 mM; metformin IC50 in 3D culture: 15.81 mM vs. 73.35 mM) (Appendix A). In addition, metformin enhanced the sensitivity of A549 and A549/PTX drug-resistant cells to PTX in 2D or 3D culture (CDI < 1 or ΔE > 0) (Appendix A). These results show that metformin has additional antiproliferative effects on PTX-resistant cells.

### 2.2. FOXO3a Regulates Stemness in NSCLC/PTX-Resistant Cells

To further test the stemness of NSCLC/PTX-resistant cells, we examined the expression of stemness-associated markers, including KLF4, OCT4, and c-MYC, hallmarks of cancer stem cells [32]. As shown in Figure 2, a striking increase in the expression of these pluripotent genes was observed in PTX-resistant cells as compared to PTX-sensitive cells (Figure 2A,B). Next, we asked if FOXO3a played a role in the development of drug resistance. Interestingly, both the protein and mRNA levels of FOXO3a were low in PTX-resistant cells (Figure 2A,C). To explore the connections between FOXO3a and stemness markers, we silenced FOXO3a in A549 cells with siRNA (Figure 2D). An obvious elevation in markers, including OCT4a, KLF4, and c-MYC, was observed in A549-FOXO3a-siRNA cells (Figure 2E). In particular, when wild-type FOXO3a (FOXO3a^WT^) or a mutant that disabled phosphorylation by Akt (FOXO3a^3A^) was ectopically expressed in PTX-resistant cells, the overexpressed FOXO3a variants blunted the increases in c-MYC and KLF4 (Figure 2F,G). These findings indicate that FOXO3a, as a tumor suppressor, prevents the cancer stemness of paclitaxel-resistant NSCLC cells.

We then used H460/PTX drug-resistant cells to generate stable cell lines infected with lentivirus expressing pCMV-GFP, wild-type FOXO3a, or mutant FOXO3a^3A^. The overexpression of FOXO3a^WT^ or FOXO3a^3A^ was confirmed at the protein level by Western blot analysis (Figure 2H). We subcutaneously implanted these three cell lines (5 × 10^6^ cells, respectively) in nude mice. As shown in Figure 2I–K, the tumors originating from FOXO3a-overexpressing cells grew very slowly (taking more than two months to grow to a palpable size), while tumors from the control (GFP) groups grew very fast. Histological examination of subcutaneous tumor sections exhibited notable features of tumor histology (Figure 2L). These results reveal that FOXO3a is a potent tumor suppressor in stemness-enriched drug-resistant cancer cells.

### 2.3. Metformin Attenuates Development of Cancer Stemness in PTX-Resistant Cells

To assess the effect of metformin on stemness, mRNA and protein levels of pluripotent genes like c-MYC, OCT4, NANOG, and NOTCH1 were analyzed by real-time PCR and Western blot. Notably, metformin strikingly repressed the expression of these CSC marker genes in A549/PTX-resistant cells (Figure 3A,B). In sphere culture assays, metformin dose-dependently reduced the numbers and diameters of A549/PTX and H460/PTX spheres (Figure 3C–E). These phenomena suggest that metformin inhibits the stemness of PTX-resistant cancer cells.

### 2.4. Metformin Upregulates FOXO3a and Induces FOXO3a Nuclear Localization

Given the inactivation of FOXO3a in PTX-resistant cells, we sought to determine whether metformin could boost the expression of FOXO3a. Our qPCR data demonstrated that metformin significantly increased FOXO3a levels in PTX-sensitive NSCLC cells. However, metformin failed to restore the level of FOXO3a in PTX-resistant cells to that in PTX-sensitive cells in spite of a moderate increase (Figure 4A). It is well-appreciated that FOXOs themselves shuttle between the nucleus and cytoplasm, which is regulated by their post-translational modifications, such as phosphorylation [33], acetylation [34], and ubiquitination [35]. To examine whether the subcellular localization of FOXO3a is involved in the development of drug resistance and regulated by metformin, we treated A549 and H460 cells and their PTX-resistant counterparts with metformin (5 mM) for 24 h and analyzed the subcellular localization of FOXO3a (Figure 4B). The data showed that FOXO3a was almost exclusively expressed outside the nucleus in PTX-resistant cells under the basal condition, and metformin promoted the nuclear localization of FOXO3a in both PTX-sensitive and -resistant cells (Figure 4C).

### 2.5. Knockdown of FOXO3a Reverses Metformin-Mediated Suppression of the Stemness in PTX-Sensitive Cells

To assess whether FOXO3a is necessary for the metformin-mediated suppression of stemness, we transfected A549 and H460 with negative control or FOXO3a siRNA and treated them with metformin (5 mM) for 24 h. Our data demonstrated that the knockdown of FOXO3a in A549 and H460 cells strikingly promoted sphere formation (Figure 4D,E). Likewise, a pronounced increase in c-MYC, which was reversed by metformin, was observed in cells with FOXO3a knockdown (Figure 4F,G). Hence, our results showed that FOXO3a mediated the inhibitory effect of metformin on cancer stemness.

### 2.6. Metformin Inhibits Stemness through AMPK/Akt/ERK-FOXO3a Signaling

As a transcription factor, FOXO3a is positively or negatively regulated by multiple signaling pathways, such as the AMPK [31], PI3K/Akt [36], and MAPK/ERK [37] pathways, and plays a pivotal role in cell proliferation, cell survival, and tumor progression. To investigate the role of FOXO3a in metformin-induced inhibition of stemness, we performed immunoblotting with total lysates of A549 and H460 PTX-resistant cells after metformin treatment. Our data revealed a remarkable increase in the expression of c-MYC and the phosphorylation of Akt and ERK, while phosphorylation of AMPK at T172 and expression of FOXO3a were dampened in PTX-resistant cells. However, administration of metformin increased the phosphorylation of AMPK and FOXO3a in a dose-dependent fashion, and reversed the increases in p-Akt, p-ERK, and c-MYC in PTX-resistant cells (Figure 5A). To determine the role of AMPK in metformin-induced upregulation of FOXO3a, we expressed a dominant negative mutant of AMPK into the cells. Our results showed that expression of the AMPK mutant (AMPK-DN) offset the effects of metformin on the levels of FOXO3a, p-AMPK, and c-MYC (Figure 5B). In addition, we used the inhibitors of Akt and MEK (i.e., MK2206 and U0126, respectively) and found that both inhibitors increased the expression of FOXO3a and downregulated c-MYC in a dose-dependent manner (Figure 5C,E). Collectively, these results suggest that the regulation of FOXO3a by metformin is co-mediated by AMPK, Akt, and ERK kinases in NSCLC/PTX-resistant cells.

### 2.7. Metformin Stabilizes FOXO3a through Ubiquitin-Proteasome Pathway

To assess if metformin stabilizes its expression, we examined the ubiquitination status of FOXO3a. Our results revealed that metformin stabilized FOXO3a and alleviated its ubiquitination in PTX-resistant cells (Figure 6A,B). Together with the data shown in Figure 2F, it is plausible that stabilized FOXO3a could dampen cancer cell stemness resulting from PTX-resistance. When the ubiquitination of endogenous FOXO3a was examined (Figure 6B), FOXO3a was eluded from being heavily ubiquitinated in the presence of metformin, MK2206, or U0126, in contrast with the control group. In addition, metformin downregulated the mRNA level in murine double minute 2 (MDM2) (Figure 6C), an E3 ligase responsible for FOXO3a degradation [38,39]. Therefore, these findings suggest that metformin increased the protein stability of FOXO3a by blocking MDM2-mediated degradation.

## 3. Discussion

Drug resistance is the bottleneck of cancer therapy, which is a great challenge in clinical medicine. A key barrier is the generation of CSCs, which exhibits great heterogeneity. Metformin was reported to sensitize CSCs to chemotherapeutic agents [40]. This provides a foundation for metformin as an adjuvant anticancer agent to improve the sensitivity of chemotherapy. In the present study, we constructed two paclitaxel-resistant NSCLC cell lines, A549/PTX and H460/PTX drug-resistant cells, and found that increased cancer stemness of these resistant cells was associated with the downregulation of FOXO3a. Our results also showed that metformin had additional antiproliferative effects and synergistic activity with PTX in the drug-resistant cells. Further, metformin upregulated FOXO3a in NSCLC/PTX-resistant cells, concurrently with the suppression of cancer stemness. Overall, our data revealed another molecular mechanism by which metformin targets cancer stemness derived from PTX-resistant NSCLC cells.

It has been reported that FOXOs are involved in chemotherapy resistance, stem cell-like characteristics, and tumor initiation [18,41]. For example, inhibition of FOXO3a facilitates long-term CSC self-renewal in ovarian cancer [42]. In the present study, we also found PTX-induced resistance inactivated FOXO3a, concurrently with increased stemness of NSCLCs. Hence, our study reinforces the notion that FOXO3a could be a noteworthy target for overcoming CSC-induced drug resistance.

Metformin, a first-line hypoglycemic agent in the treatment of type 2 diabetes, has anticancer activity [21,43]. It has been reported that metformin inhibits the signaling pathways associated with cancer cell self-renewal, such as the hedgehog (Hh) [44], Wnt [45,46], transforming growth factor beta (TGF-β) [47,48], and metabolic pathways [49,50]. In our study, we observed that silencing FOXO3a reversed the metformin-induced suppression of sphere formation in 3D cultures and upregulated the expression of c-MYC in PTX-sensitive cells, whereas overexpression of FOXO3a in the PTX-resistant cells generated mimicked the effect of metformin. Therefore, our study adds another mechanism underlying metformin’s inhibition of CSCs through a FOXO3a-dependent pathway.

How metformin regulates the FOXO3a-mediated suppression of stemness is an intriguing question. We first assessed the role of AMPK, as previous studies have demonstrated that AMPK directly or indirectly modulates FOXO3a transcriptional activity [51,52,53]. We found that the abundance of AMPK was reduced in PTX-resistant cells. When a dominant negative mutant of AMPK was ectopically expressed in these cells, the expression of c-MYC was enhanced and resistant to metformin, suggesting that the AMPK–FOXO axis is recruited by metformin to suppress cancer stemness in NSCLC/PTX-resistant cells. A recent study demonstrated that AMPK regulates FOXO3a activity only when the latter has already been translocated to the nucleus, sensing a lack of both growth factors and energy, and that AMPK activates the nuclear fraction of FOXO3a [31]. The authors pointed out that AMPK-induced phosphorylation of FOXO3a does not affect the subcellular localization of FOXO3a. It is possible that metformin mobilizes cytoplasmic FOXO3a to the nucleus by inhibiting pathways other than AMPK, such as Akt and ERK, and that AMPK then phosphorylates FOXO3a in the nucleus or that AMPK-dependent and -independent mechanisms work simultaneously. Indeed, we observed that metformin enhanced the nuclear translocation of FOXO3a. Our next study will assess if this process is blocked by the dominant negative mutant of AMPK.

In tumor cells, FOXO3a is phosphorylated by Akt on three conserved amino acid residues (T32, S253, and S315), so that phosphorylated FOXO is sequestered in the cytoplasm by binding to 14-3-3 [54]. Another regulatory mechanism entails the oncogenic kinase ERK [55,56], which downregulates FOXO3a by phosphorylating FOXO3a at Ser 294, Ser 344, and Ser 425 [37]. In this study, we found that the reduced expression of FOXO3a in PTX-resistant cells was associated with increases in Akt and EKR activation and offset by Akt and ERK inhibitors. Both kinases were inactivated by increased doses of metformin. Taken together, these results suggest that the effects of metformin on FOXO3a function are simultaneous with the activation of AMPK and inhibition of Akt and ERK kinases, leading to the inhibition of cancer stemness.

Several studies have illustrated that FOXO3a is regulated by proteosome-mediated degradation. First, ERK has been shown to induce FOXO3a phosphorylation, leading to its degradation via the MDM2-mediated ubiquitin–proteasome pathway [37]. Second, AMPK can inhibit MDM2 by sequestering it the cytoplasm through Akt, leading to the stabilization of FOXO3a [38]. Third, tongue cancer resistance-associated protein (TCRP1) can induce the phosphorylation of Akt and FOXO3a, thereby blocking FOXO3a’s nuclear localization to favor FOXO3a’s ubiquitination in the cytoplasm [57]. One study revealed that active Akt enables MDM2 phosphorylation at Ser166, which in turn promotes FOXO3a degradation [38]. Therefore, the inhibition of Akt and ERK is able to mitigate the degradation of FOXO3a. In line with these studies, we found that metformin as well as inhibitors of Akt and ERK suppress FOXO3a degradation, possibly through the inhibition of MDM2.

How does increased expression of FOXO3a in response to metformin benefit cancer prevention and treatment? One study showed that FOXO3a plays a role in the DNA damage response [58]. Thus, in this scenario, FOXO3a associates with KAT5/Tip60 to activate ATM, resulting in a DNA damage checkpoint, while Notch1 competes with FOXO3a to bind KAT5/Tip60, preventing the activation of ATM [58]. Another report showed that metformin activates FOXO3a to cause the inhibition of c-MYC and STAT3, which control the expression of PD-L1 [59]. Therefore, via the activation of FOXO3a, metformin can enhance the efficacy of anti-PD-L1 immunotherapy.

In summary, our study illustrates the mechanism by which metformin exerts inhibitory effects on cancer stemness in PTX-resistant cells. It activates AMPK and inhibits Akt and ERK, all of which funnel to FOXO3a and eventually lead to the inhibition of stemness in NSCLC cells. While our study highlights the potential of metformin in the treatment of cancer stemness and chemotherapeutic resistance, the detailed mechanisms underlying the route from metformin to FOXO3a are still at infant stages. Our next questions are whether the action of metformin on Akt and ERK is dependent on AMPK, and if so, what its mechanism is.

## 4. Materials and Methods

### 4.1. Reagents

BCA kit, penicillin/streptomycin mix, and puromycin were purchased from Beijing Solarbio Company (Beijing, China). Cell counting kit (CCK-8) and SYBR Green Master Mix were purchased from Yeasen Biotechnology Co., Ltd. (Shanghai, China). Recombinant human epidermal growth factor (EGF) was from Sino Biological, Inc. (Beijing, China). Recombinant human insulin-like growth factor (IGF-1) was from PeproTech, Inc. (Rocky Hill, NJ, USA). Annexin V-FITC Apoptosis Detection Kit was procured from Boster, Inc. (Wuhan, China). Akt-specific inhibitor (MK-2206) was from MedChem Express (Monmouth, NJ, USA). MEK-specific inhibitor (U0126) was from GLPBIO (Montclair, CA, USA). Lipofectamine^®^ 3000, B27 supplement, cell culture medium, and fetal bovine serum were from Thermo Fisher Scientific, Inc. (Waltham, MA, USA). All antibodies with catalog numbers and manufacturers are listed in Table 1.

### 4.2. Cell Culture

Cell lines used in this study were cultured in DMEM supplemented with 10% fetal bovine serum and 1% penicillin/streptomycin at 37 °C in a humidified atmosphere of 5% CO_2_. Lung adenocarcinoma cell lines A549 and H460 were purchased from American Type Culture Collection (ATCC, Manassas, VA, USA). Paclitaxel (PTX)-resistant cell lines A549/PTX and H460/PTX were derived from their parental cells by incubation with a stepwise increase in paclitaxel concentrations [60]. Briefly, A549 and H460 cells were cultured in DMEM complete medium supplemented with an initial PTX concentration of 10 nM; the cells were then transitioned to a PTX-free medium upon elimination of non-viable cells. After the cells recovered and entered the logarithmic growth phase, a stepwise elevation in PTX concentration was added to the medium, and then to a PTX-free medium after dead cells were removed for recovery. The process was repeated over a span of 4–6 months.

### 4.3. Cell Viability Assays

Cells were seeded into a 96-well plate at a density of 2000 cells/well for 3 days. The assay was conducted using a CCK-8 kit according to the manufacturer’s protocol. In brief, the CCK8 solution was added onto the culture and incubated for 2 h. Absorbance at 450 nm was measured using an auto-microplate reader (SpectraMax Paradigm Molecular Devices, Silicon Valley, CA, USA). The percentage of inhibition was calculated using the following equation: inhibition percentage = (1 − OD_experiment_/OD_control_) × 100. The IC_50_ value was determined by plotting the inhibition percentage values. The resistance index (RI) was calculated using the following equation: RI = half-maximal inhibitory concentration (IC_50_)_PTX-resistant cells_ /(IC_50_)_parental cells_.

### 4.4. Tests for Synergism and Interaction

The coefficient of drug interaction (CDI) was employed to assess the synergistic inhibitory effect of drug combinations, as described previously [61,62].The CDI was computed using the formula CDI = AB/(A × B). In this equation, AB represents the ratio of the combination groups to the control group, while A and B denote the ratio of the single-agent group to the control group. Drug interaction is classified as synergistic, additive, or antagonistic based on whether CDI is less than, equal to, or greater than 1. A CDI value below 0.7 indicates a significant synergistic effect.

The Bliss Independence (BI) theory was applied to determine the synergy between paclitaxel and metformin used to treat PTX-sensitive and -resistant NSCLC cells. The BI theory for the two agents is expressed by the equation E_pre_ = E_A_ + E_B_ − (E_A_ × E_B_), where E is the fractional effect and A and B are the doses of the two compounds in a combination experiment. The difference (ΔE) between the predicted percentage of growth (E_pre_) and experimental observed percentage of growth (E_exp_) defines the interaction of the combination of each drug. If the combination effect is higher than the predicted value, the interaction is synergistic; if it is lower than the predicted value, the interaction is antagonistic; otherwise, the effect is additive and there is no interaction [63].

### 4.5. Enrichment of Stem Cell-like Spheres

Cells were digested with trypsin/EDTA (0.25%) and washed with PBS (BI; Biological Industries) twice. Subsequently, cells were seeded at a density of 1 × 10^4^ cells/well in six-well low-attachment plates and cultured in stem cell-conditioned culture medium (DMEM/F12 medium containing 2% B27 supplement, EGF (20 ng/mL) and IGF-1 (20 ng/mL)) [64]. Culture medium was half-replaced every other day. When the diameters of spheres reached 90–100 µm after approximately 5 days, the spheres were collected, trypsinized into a single cell suspension, and resuspended in fresh medium for serial subcultivation every 3 days. The morphology of the spheres was recorded with an inverted microscope (Olympus, Tokyo, Japan) every 3 days. The total number and diameter of spheres (>50 µm) in 5 typical fields in each well were counted, and the mean values were calculated accordingly.

### 4.6. Flow Cytometric Analysis

Cell apoptosis was assessed with the Annexin V-FITC Apoptosis Detection Kit according to the manufacturer’s instructions. In brief, 2.5 × 10^5^ cells were collected, washed twice with PBS, resuspended in 500 μL binding buffer, and incubated with 5 μL Annexin V-FITC and 5 μL propidium iodide (PI) for 10 min before analysis on a Cytoflex flow cytometer (Beckman Coulter, Brea, CA, USA).

### 4.7. Virus Production and Transfection

The lentiviral vectors bearing GFP-puro, FOXO3a^WT^, or FOXO3a^3A^ (T32A, S253A, and S315A) plasmids were constructed using vectors provided by Beijing Tsingke Technology Co., Ltd. (Beijing, China). A549/PTX and H460/PTX drug-resistant cells were infected with lentivirus expressing wild-type or mutant FOXO3a and selected with puromycin (1 µg/mL). AMPK dominant negative mutant (AMPK-DN) plasmid or empty vector (EV) plasmid were transfected into A549/PTX drug-resistant cells as described [65]. FOXO3a siRNA (si-FOXO3a) and its negative control (si-NC) were designed from OriGene Technologies, Inc. (Rockville, MD, USA). A549 and H460 cells were transfected with 200 pmol si-FOXO3a and si-NC using Lipofectamine^®®^ 3000 (Cat. no. 2435467, Invitrogen, Carlsbad, CA, USA). Transfected cells were incubated for an additional 24 h before metformin treatment. The siRNA sequences were as follows:

siRNA-FOXO3a control (NC): CGUUAAUCGCGUAUAAUACGCGUAT;

siRNA-FOXO3a#1: CAAAUGUCACUAAAGGGUUUAGUTT;

siRNA-FOXO3a#2: AGAGGCAAUAGCAUACAAACUGATT.

### 4.8. Real-Time PCR

Primers for CSC markers (c-MYC, OCT4, NANOG and NOTCH1) and FOXO3a were purchased for qPCR from Tsingke Biotechnology Co., Ltd. (Beijing, China). All primer sequences are listed in Table 2. Cells were cultured in 60 mm dishes treated with metformin to a final concentration of 10 mM for 24 h. Total RNA was prepared, and first-strand cDNA was synthesized with 1 μg of RNA as a template from each sample using an iScript cDNA synthesis kit from TransGen Biotechnology Co., Ltd. (Beijing, China) (Cat: AE311-03). qPCR was performed using the SYBR Green Master Mix (Cat: 11202ES08, Yeasen). Ct values were normalized with Ct values of GAPDH as an internal control.

### 4.9. Western Blotting

Cells were harvested and lysed on ice with RIPA buffer containing protease inhibitors [66]. The cell lysate was centrifuged at 13,000 rpm in an Eppendorf centrifuge for 15 min at 4 °C and protein concentration was measured by BCA. The supernatant (20 µg) was subjected to SDS-PAGE and electrophoretically transferred onto polyvinylidene fluoride (PVDF) membrane. The PVDF membrane was then blocked in Tris-buffered saline + 0.1% Tween-20 (TBST) containing 5% non-fat milk for 1 h at room temperature, incubated with antibodies at 4 °C overnight, and then washed 3 times with TBST for 10 min each. Then, the membranes were incubated with HRP-conjugated second antibodies at room temperature for 2 h, followed by washing with TBST for 3 times. Specific proteins were visualized by enhanced chemiluminescence (ECL).

### 4.10. Xenograft Tumor Model

Male BALB/c nude mice at 4 weeks old were purchased from GemPharmatech Co., Ltd. (Nanjing, Jiangsu, China), acclimatized in the animal center for one week, and randomly subdivided into 3 groups (n = 6): GFP-Control, FOXO3a-WT, and FOXO3a-3A groups. The cells (5 × 10^6^/100 μL in PBS) were injected subcutaneously into the flanks of nude mice. When tumors were palpable, the volumes were measured every other day according to the formula of length × width^2^ × 0.5, where width is the shorter diameter and length is the longer diameter. Finally, the mice were sacrificed at day 20 (for H460/PTX-GFP groups) or day 68 (for H460/PTX FOXO3a-WT and FOXO3a-3A groups), and tumors were collected for measurement of tumor weight and morphological analysis. The animal protocol was approved by the Animal Use and Care Committee of Nanchang University (NCDXSYDWFL-2015097).

### 4.11. Hematoxylin and Eosin (H&E) Staining

H&E staining was performed as described by Zhang et al. [67]. In brief, the tissues were fixed in 4% paraformaldehyde overnight at 4 °C, processed, and embedded in paraffin. The embedded tissues were cut into 4 μm slides, dewaxed, and stained with H&E. At last, images at different magnifications were captured by microscopic photography.

### 4.12. Statistical Analyses

Data analysis was performed using the statistical software SPSS 19.0. For continuous variables, data were expressed as mean ± standard deviation (mean ± SD). Student’s *t*-test was used to compare mean differences between two groups. Significant difference was defined as *p* < 0.05 (* *p* < 0.05, ** *p* < 0.01, *** *p* < 0.001).

## Figures and Tables

**Figure 1 ijms-24-16611-f001:**
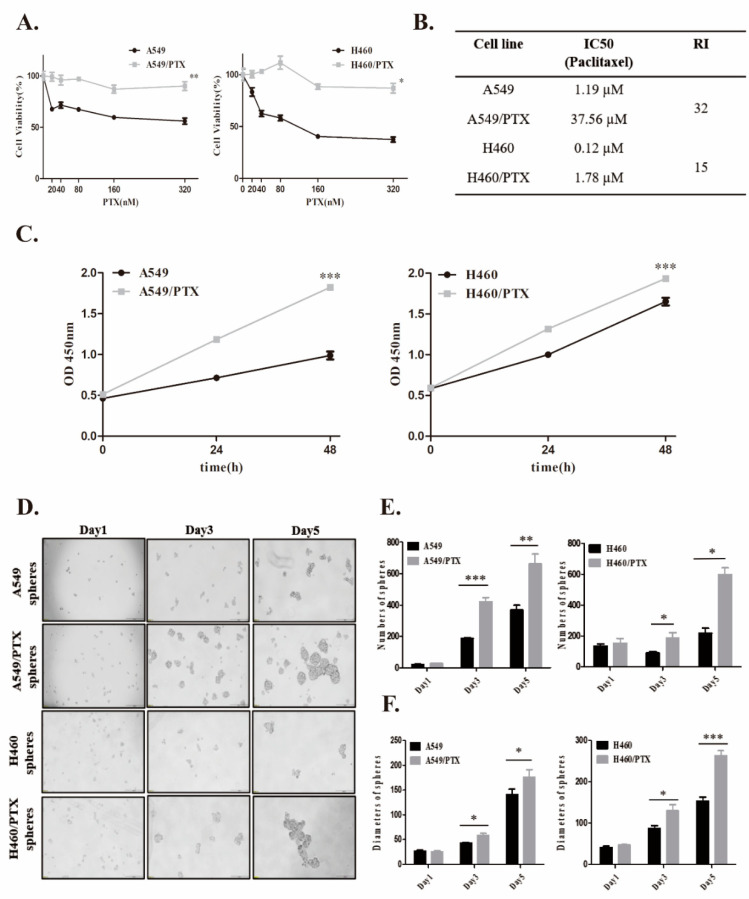
The stemness of NSCLC/PTX-resistant cells. (**A**) Paclitaxel-resistant cells were established by intermittent incubation of A549 and H460 cells with increasing concentrations of paclitaxel. The cells were treated with paclitaxel (20–320 nM) for 48 h and viabilities were calculated as mean ± SD (n = 3). (**B**) IC_50_ values were calculated by Cell counting Kit-8 (CCK-8) assay. RI: resistance index. (**C**) Cell viabilities of parental and PTX-resistant cells were examined using CCK-8 kit (mean ± SD, n = 6). Statistical analysis was performed using Student’s *t*-test. (**D**) Typical bright field images of cells cultured in stem cell-conditioned medium on days 1, 3, and 5. Scale bar, 200 µm. (**E**) Numbers and (**F**) average diameters of suspended spheres on days 1, 3, and 5. Data are expressed as mean ± SD (n = 5), * *p* < 0.05, ** *p* < 0.01, *** *p* < 0.001.

**Figure 2 ijms-24-16611-f002:**
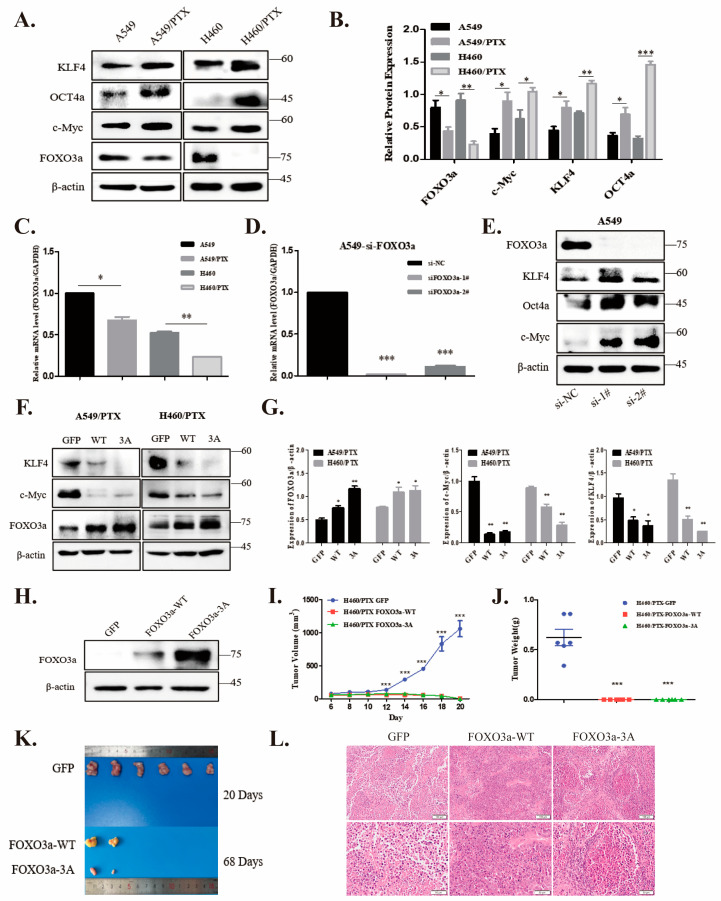
FOXO3a regulates stemness of NSCLC/PTX-resistant cells. (**A**,**B**) Expression of stemness markers and FOXO3a were analyzed by Western blot with antibodies as indicated (**A**) and scan densitometric units were plotted (**B**). (**C**) The mRNA levels of FOXO3a were assessed by qPCR. (**D**,**E**) Efficiency of knockdown by siRNA-FOXO3a was validated using Q-PCR and immunoblotting with antibodies as indicated. (**F**,**G**) Lysates of A549/PTX and H460/PTX cells transfected with pCMV-GFP, wild-type FOXO3a, or mutant FOXO3a^3A^ were blotted with antibodies as indicated. (**H**) Overexpression of FOXO3a suppresses tumor growth in nude mice. Stable cell lines of H460/PTX containing pCMV-GFP, wild-type FOXO3a, or mutant FOXO3a^3A^ were examined by immunoblotting. (**I**) Subcutaneous implantation of BALB/C nude mice was performed with the cells as indicated in (**H**). Each cell line was injected into 6 animals. Tumor growth was monitored every other day. (**J**) Weight of tumors. The inoculation rate of tumors with cells expressing FOXO3a was very low. (**K**) Image of tumors. (**L**) Hematoxylin and eosin (H&E) staining of tumor tissue sections obtained from xenograft tumor models. Representative images are shown (magnification ×40, ×200). All experimental data are expressed as mean ± SD (for cell-based assay: n = 3, animal assay: n = 6), * *p* < 0.05, ** *p* < 0.01, *** *p* < 0.001.

**Figure 3 ijms-24-16611-f003:**
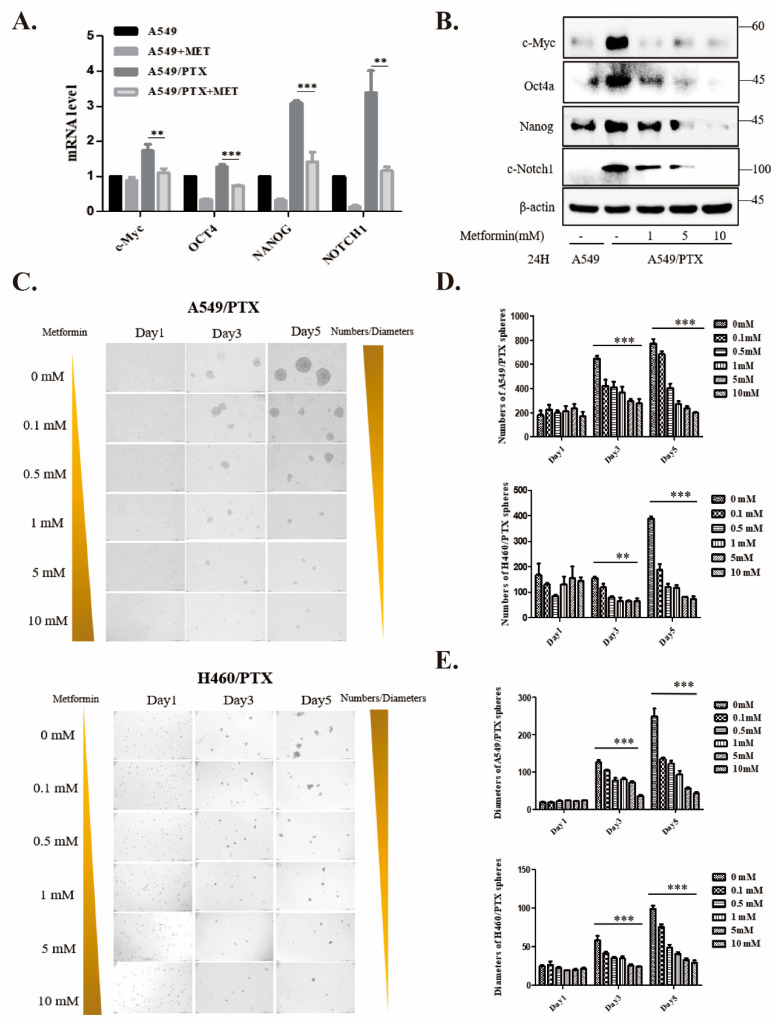
Metformin reduces stemness properties in NSCLC/PTX-resistant cells. (**A**,**B**) qPCR (**A**) and Western blot (**B**) showing the relative expression of cancer stem cell markers in A549 and A549/PTX drug-resistant cells in response to metformin treatment. GAPDH or β-actin was used as an internal control. (**C**) Typical bright-field images of A549/PTX and H460/PTX spheroids treated with 0–10 mM metformin. Scale bar, 200 µm. (**D**,**E**) Average numbers (**D**) and diameters (**E**) of suspended spheres on days 1, 3, and 5. Data are expressed as mean ± SD (n = 3), ** *p* < 0.01, *** *p* < 0.001.

**Figure 4 ijms-24-16611-f004:**
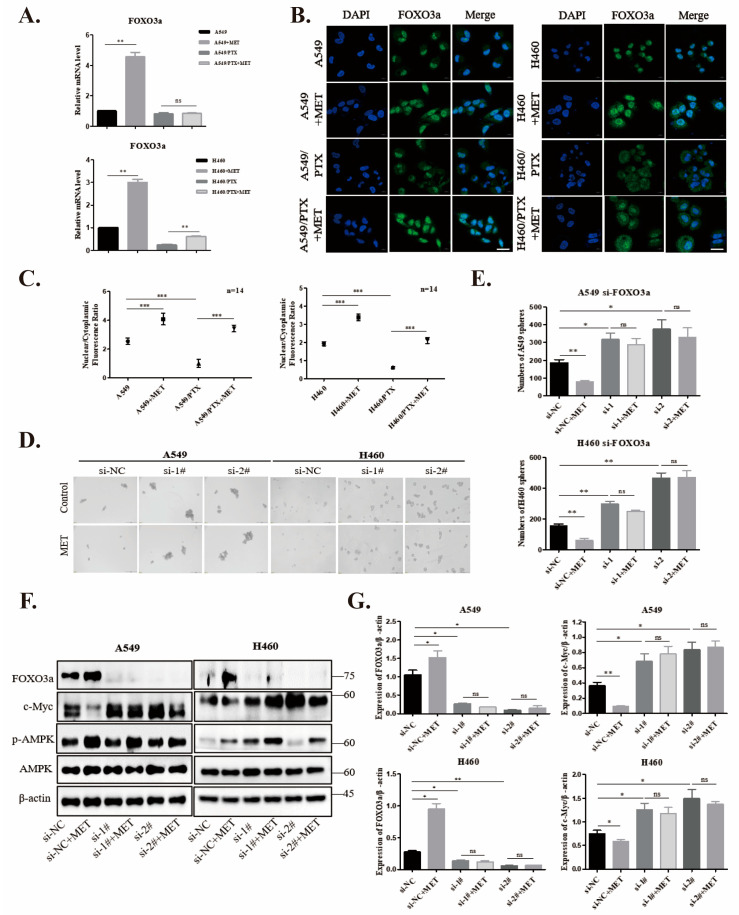
FOXO3a mediates the inhibitory effect of metformin on PTX-induced stemness. (**A**) mRNA levels of FOXO3a were assessed using qPCR in parental and PTX-resistant cells after treatment with or without metformin (10 mM) for 24 h. (**B**) The subcellular localization of FOXO3a was labeled with antibody against FOXO3a followed by Alexa Fluor 488-conjugated secondary antibody and visualized under fluorescence microscopy. DAPI was used to stain nuclei. Scale bar: 10 µm. (**C**) Nuclear localization of FOXO3a was assessed in cells as a ratio of nuclear to cytoplasmic fluorescence using Image J software V1.8.0. Data are means ± SD from n = 14 fluorescent cells analyzed. (**D**) A549 and H460 cells were transfected with FOXO3a-siRNA or control-siRNA, grown in 3D sphere-cultures for 3 days, and then incubated with or without metformin (5 mM) for 24 h. The cells were visualized under a bright-field microscope. (**E**) The histogram shows the numbers of spheres of A549 and H460-FOXO3a-siRNA cells vs. control-siRNA. (**F**,**G**) Total lysates of metformin-treated A549 and H460 transfected with control-siRNA or FOXO3a-siRNA were analyzed by Western blot with antibodies as indicated. Data are expressed as mean ± SD (n = 3); ns means no significance, * *p* < 0.05, ** *p* < 0.01, *** *p* < 0.001.

**Figure 5 ijms-24-16611-f005:**
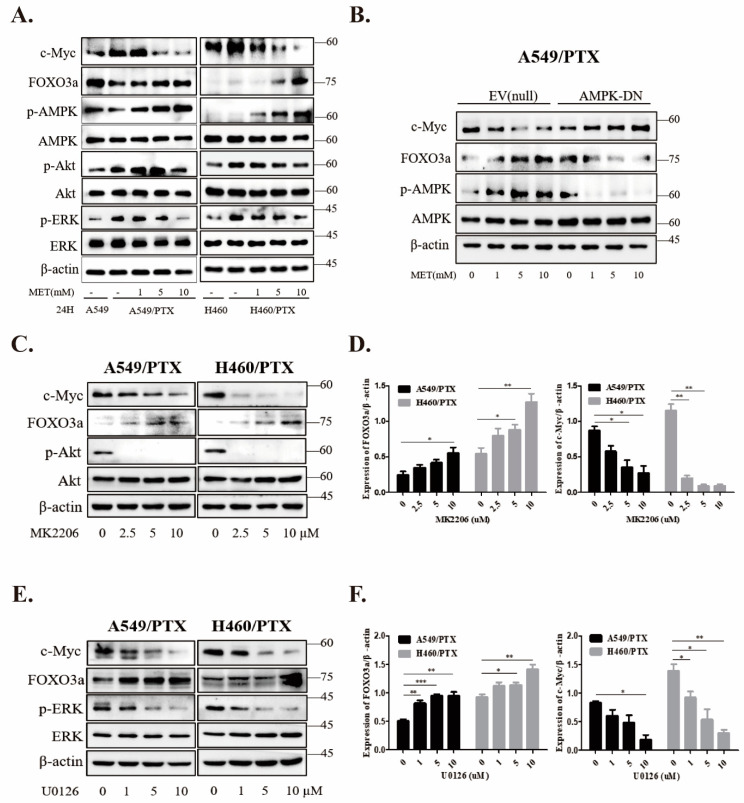
The role of AMPK in metformin regulation of FOXO3a signaling in NSCLC/PTX-resistant cells. (**A**) PTX-resistant cells were treated with metformin at increasing doses for 24 h and cell lysates were prepared for immunoblotting with antibodies as indicated, while PTX-sensitive cells without metformin were used as controls. (**B**) A549/PTX cells were transfected with AMPK dominant negative mutant (AMPK-DN) or empty vector (EV) and treated with different dosages of metformin for 24 h. Immunoblotting was performed as in (**A**). (**C**,**D**) Cells were treated with DMSO or MK2206 for 24 h. (**E**,**F**) Cells were treated with DMSO or U0126 for 24 h. Data are expressed as mean ± SD (n = 3), * *p* < 0.05, ** *p* < 0.01, *** *p* < 0.001.

**Figure 6 ijms-24-16611-f006:**
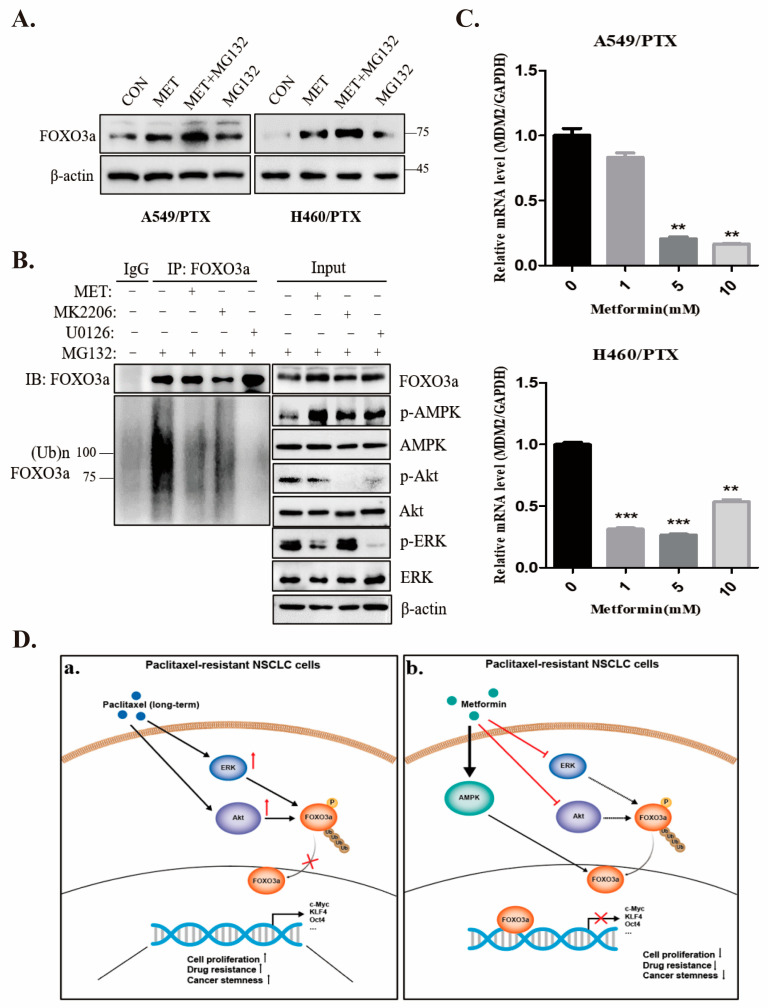
Metformin stabilizes FOXO3a through inhibiting ubiquitination and degradation. (**A**) A549/PTX and H460/PTX-resistant cells were treated with metformin (10 mM) or MG132 (10 µM) alone or in combination for 24 h and cell lysates were blotted with the indicated antibodies. (**B**) A549/PTX-resistant cells were treated with metformin (10 mM), MK2206 (10 µM), or U0126 (10 µM) for 24 h and 8 h. Before the end of the experiment, MG132 was added. Cell lysates were subjected to immunoprecipitation and immunoblotting with the indicated antibodies. (**C**) mRNA levels in MDM2 were examined by qPCR. PTX-resistant cells were treated with or without metformin (0, 1, 5 and 10 mM) for 24 h. Data are expressed as mean ± SD (n = 3), ** *p* < 0.01, *** *p* < 0.001. (**D**) The mechanisms by which metformin suppresses PTX-induced stemness of NSCLCs. a. Long-term treatment with paclitaxel leads to activation of Akt and ERK in NSCLC cells, resulting in inactivation of FOXO3a through phosphorylation and ubiquitination. b. Metformin exerts inhibitory effects via multiple mechanisms: (1) activation of AMPK, (2) inhibition of Akt and ERK. As a result, increased FOXO3a suppresses the stemness of paclitaxel-resistant NSCLC cells. Arrows indicate activation, cross symbols indicate inhibition.

**Table 1 ijms-24-16611-t001:** List of antibodies used in this article.

Target	Source	Catalog No.
p-Akt (Ser473)	Cell Signaling Tech. (Danvers, MA, USA)	4060S
Akt	Cell Signaling Tech.	13038
p-AMPK (Thr172)	Cell Signaling Tech.	8208
AMPK	Cell Signaling Tech.	5832
p-ERK (Thr202/Tyr204)	Cell Signaling Tech.	4370
ERK	Cell Signaling Tech.	4695S
Nanog	Cell Signaling Tech.	4903P
Cleaved-Notch1	Cell Signaling Tech.	2421S
Oct4a	Cell Signaling Tech.	2840P
KLF4	Cell Signaling Tech.	4038P
c-Myc	Cell Signaling Tech.	5605P
FOXO3a	Proteintech (Wuhan, China)	10849-1-AP
Ubiquitin	Proteintech (Wuhan, China)	10201-2-AP
β-actin	OriGene (Rockville, MD, USA)	TA811000

**Table 2 ijms-24-16611-t002:** Sequences of primers used for quantitative real-time PCR assay.

Gene Names	Primers (5′–3′)
GAPDH	F: ACGGGAAGCTCACTGGCATGG
	R: GGTCCACCACCCTGTTGCTGTA
Oct-4	F: CTTCTCGCCCCCTCCAGGT
	R: AAATAGAACCCCCAGGGTGAGC
Nanog	F: TGAACCTCAGCTACAAACAG
	R: TGGTGGTAGGAAGAGTAAAG
Notch1	F: TCAGCGGGATCCACTGTGAG
	R: ACACAGGCAGGTGAACGAGTTG
c-MYC	F: GGACGACGAGACCTTCATCAA
	R: CCAGCTTCTCTGAGACGAGCTT
FOXO3a	F: CGGACAAACGGCTCACTCT
	R: GGACCCGCATGAATCGACTAT
MDM2	F: GAATCATCGGACTCAGGTACATC
	R: TCTGTCTCACTAATTGCTCTCCT

## Data Availability

All data generated during this study are included in this published article and the datasets described in this study can be acquired from the corresponding author upon request.

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
