# Peer review of "Metformin Suppresses Stemness of Non-Small-Cell Lung Cancer Induced by Paclitaxel through FOXO3a"

_ijms, 2023, doi:10.3390/ijms242316611_

Round 1

Reviewer 1 Report

Comments and Suggestions for Authors

1.     This manuscript is interesting and well-done.

2.     The strength of this article is well organized for readers to understand for anti-cancer effect of metformin in non-small cell lung cancer.

3.     Moreover current study proposed that metformin could be therapeutic approach to suppress non-small cell lung cancer.

4. In figure 1E (right, Day5), 1B (OCT4a), 1G (WT), 3A (NOTCH1), 4G, 5F, It's just my opinion, standard deviation seems a little high. Could you tell me reason of high STDEV?

5. Metforminis known as antidiabetics. However it was often used as a cure for cancer. Current study proposed that metformin is suppressed to NSCLC via paclitaxel mediated FOXO3a down-modulated. I recommend adding a brief explanation in 'discussion' about transcriptionaly mechanism between FOXO3a and Notch1 under treatment metformin.

6. It's just my opinion, this manuscript was accepted to be enough after english language and style are minor spell check.

Comments on the Quality of English Language

1.     It's just my opinion, this manuscript was accepted to be enough after english language and style are minor spell check.

Author Response

We thank the reviewer 1 for the positive comments on our work and suggestions.

Comments 1: In figure 1E (right, Day5), 1B (OCT4a), 1G (WT), 3A (NOTCH1), 4G, 5F, It's just my opinion, standard deviation seems a little high. Could you tell me reason of high STDEV?

Response 1: Thanks for the comment. High standard deviation values were caused by relatively small sample sizes. Thus, in the revised version, we increased the sample size.

Comments 2: ……I recommend adding a brief explanation in 'discussion' about transcriptionally mechanism between FOXO3a and Notch1 under treatment metformin.

Response 2: Thanks for this suggestion. We have added a brief explanation in the Discussion. The addition can be found on page 14, paragraph 4, lines 368-371.

Comments 3: It's just my opinion, this manuscript was accepted to be enough after English language and style are minor spell check.

Response 3: Thanks. We have carefully checked and edited.

Reviewer 2 Report

Comments and Suggestions for Authors

This study aims to analyze the mechanism of metformin in FOXO3 inhibition of paclitaxel resistance-induced cancer stem cell phenotype in non-small cell lung cancer (NSCLC) cell lines.

The article is well written and the methodology is well described and adequate. The results described in the article are novel and of interest in the search for new therapeutic strategies in paclitaxel-resistant NSCLC.

However, additional experiments on methfomine-induced cell death in different cell lines would be desirable.

Questions for the authors

Figure 1. A.

Although viability assays indicate that resistant cells have a higher proliferative capacity, cell death assays would be necessary to determine that they are also more resistant to paclitaxel-induced cell death.

Could they provide cell death data by studying caspase activation or PARP cleavage by western blot or AnnexinV assay?

Figure 1.D. Sphere formation is convincing in A549/PTX cells, however, it is more doubtful in H460/PTX cells. ¿ could you show better images?

Figure 2. The legend of some of the sections is too small. The authors should consider increasing the font size.

Figure 2.J. It would be necessary to show histological images of the tumor lesions.

Regarding the animal assay, could the authors explain the advantages of performing the assays with the treatments in the animal model with respect to performing them in the cell lines used in the article?

Assays on cell lines would provide a better understanding of the mechanisms of cell death induced by the treatments.

The images in Figure 4B are of low quality and it is difficult to appreciate the nuclear translocation of FOXO3.

Author Response

The comments from the reviewer 2 were in general positive. However, it was raised that additional experiments on metformin-induced cell death in different cell lines would be desirable.

Comments 1: Figure 1.A. Although viability assays indicate that resistant cells have a higher proliferative capacity, cell death assays would be necessary to determine that they are also more resistant to paclitaxel-induced cell death. Could they provide cell death data by studying caspase activation or PARP cleavage by western blot or Annexin V assay?

Response 1: Thanks for the thoughtful suggestions. We have performed Annexin V assay to evaluate the cytotoxicity of paclitaxel in A549 and A549/PTX drug resistant cells, and found that A549/PTX-resistant cells are more resistant to paclitaxel-induced cell death than A549 cells. The pattern has been added as the Figure S1 in the revised supporting information.

Comments 2: Figure 1.D. Sphere formation is convincing in A549/PTX cells, however, it is more doubtful in H460/PTX cells. Could you show better images?

Response 2: We totally understand this concern. Now we presented better images of H460/PTX spheres in figure 1D and 4D.

Comments 3: Figure 2. The legend of some of the sections is too small. The authors should consider increasing the font size.

Response 3: Thanks. It is done. 

Comments 4: Figure 2.J. It would be necessary to show histological images of the tumor lesions.

Regarding the animal assay, could the authors explain the advantages of performing the assays with the treatments in the animal model with respect to performing them in the cell lines used in the article? Assays on cell lines would provide a better understanding of the mechanisms of cell death induced by the treatments.

Response 4: Thanks. New experiments as suggested were performed with Hematoxylin and Eosin (H&E) Staining to visualize the histology of the subcutaneous tumor lesions, and now presented in the Figure 2L.

We agreed that assays on cell lines is better for mechanistic studies, while animal models provide supporting information on biological relevance. For example, we are not sure if changes in cell behavior have biological relevance if we do not have in vivo data using animal models.  However, we are aware that Xenograft models are not the same as naturally occurring tumor in human.

Comments 5: The images in Figure 4B are of low quality and it is difficult to appreciate the nuclear translocation of FOXO3.

Response 5: Thanks for this criticism. We have reperformed the immunofluorescence assay to analyze the subcellular localization of FOXO3a under metformin treatment or basal condition in PTX-sensitive and resistant cells, and found that metformin promoted nuclear localization of FOXO3a in both cell lines. The new images and data can be found in figure 4B and 4C.

Reviewer 3 Report

Comments and Suggestions for Authors

This is an overall well-conducted study of induced resistance to PTX-chemotherapy drugs in NSCLC cell lines. The findings in this study are important for overcoming resistance to PTX- treatment.

The strengths of this study include

1. There is a comprehensive and well-thought-out series of experiments to address several aspects of cancer growth and biology.

2. Experiments were well-described and data was presented that overall support the main thesis of the manuscript.

3. A new potential mechanism was identified in which metformin has great potential to inhibit PTX-resistant lung cancer stem cells.

However, some questions arise concerning the potential for improvement in the manuscript. These include

1.      Other studies designed to evaluate resistance have been conducted in the introduction and discussion. However, little detail is given in either section. Specifically, little detail is given concerning immunotherapy as a new therapy for NCSLC also in combination with chemotherapy, in the introduction (lane number 32).

2.      Figures are hard to see and identify details especially the legend and the text of the axes.

3.      Calculate the IC50 for all the treatments and show the combination index values to understand the synergistic effect in at list one cell line 2D resistant vs. sensitive, 3D resistant vs. sensitive.  Conduct the Bliss or the Chou-Talaly model.

Comments on the Quality of English Language

After some editing for punctuation and grammar, the final form will be easily readable.

Author Response

We thank the reviewer 3 for his/her enthusiasm about our work.

Comments 1: Other studies designed to evaluate resistance have been conducted in the introduction and discussion. However, little detail is given in either section. Specifically, little detail is given concerning immunotherapy as a new therapy for NCSLC also in combination with chemotherapy, in the introduction (lane number 32).

Response 1: According to suggestions, we have expanded introduction regarding the combination of immunotherapy and chemotherapy in NSCLC in the ‘Introduction’ (page 1, paragraph 2, lines 32-34). We also discussed the role of metformin in sensitizing immunotherapy by activating FOXO3a in the ‘Discussion’. Corresponding paragraphs were added on page 14, paragraph 4, lines 371-374.

Comments 2: Figures are hard to see and identify details especially the legend and the text of the axes.

Response 2: Thanks for the comment. We have enlarged the legend and text.

Comments 3: Calculate the IC50 for all the treatments and show the combination index values to understand the synergistic effect in at list one cell line 2D resistant vs. sensitive, 3D resistant vs. sensitive. Conduct the Bliss or the Chou-Talaly model.

Response 3: Thanks again your valuable comment. We have performed cell viability assays using CCK-8 kits to calculate the IC50 values for all the treatments, and found that A549/PTX drug-resistant cells are more sensitive to metformin but resistant to PTX. In addition, we have applied Bliss Independence (BI) theory to determine synergy between paclitaxel and metformin that were used to treat A549 and A549/PTX drug-resistant cells. The pattern has been added as the Figure S2 in the revised supporting information.

Comments 4: Comments on the Quality of English Language. After some editing for punctuation and grammar, the final form will be easily readable.

Response 4: Done.

Round 2

Reviewer 2 Report

Comments and Suggestions for Authors

I thank the authors for their efforts to answer the questions that were not clear in the previous version.

Reviewer 3 Report

Comments and Suggestions for Authors

Dear authors,

I just wanted to reach out and let you know that all of your requests have been completed. I wanted to take a moment to commend you on your good work. It's clear that you put a lot of effort and dedication into your project.